# Vitamin D3 Enriches Ceramide Content in Exosomes Released by Embryonic Hippocampal Cells

**DOI:** 10.3390/ijms22179287

**Published:** 2021-08-27

**Authors:** Carmela Conte, Samuela Cataldi, Cataldo Arcuri, Alessandra Mirarchi, Andrea Lazzarini, Mercedes Garcia-Gil, Tommaso Beccari, Francesco Curcio, Elisabetta Albi

**Affiliations:** 1Department of Pharmaceutical Sciences, University of Perugia, 06126 Perugia, Italy; carmela.conte@unipg.it (C.C.); samuela.cataldi@unipg.it (S.C.); tommaso.beccari@unipg.it (T.B.); 2Department of Medicine and Surgery, University of Perugia, 06126 Perugia, Italy; cataldo.arcuri@unipg.it (C.A.); alemirarchi@libero.it (A.M.); 3Crabion S.R.L., 06100 Perugia, Italy; andrylazza@gmail.com; 4Department of Biology, Interdepartmental Research Center “Nutraceuticals and Food for Health”, University of Pisa, 56127 Pisa, Italy; mercedes.garcia@unipi.it; 5Department of Medicine, Institute of Clinical Pathology, University of Udine, 33100 Udine, Italy; francesco.curcio@uniud.it

**Keywords:** ceramide, exosome, sphingomyelin, embryonic cell differentiation, vitamin D

## Abstract

The release of exosomes can lead to cell–cell communication. Nutrients such as vitamin D3 and sphingolipids have important roles in many cellular functions, including proliferation, differentiation, senescence, and cancer. However, the specific composition of sphingolipids in exosomes and their changes induced by vitamin D3 treatment have not been elucidated. Here, we initially observed neutral sphingomyelinase and vitamin D receptors in exosomes released from HN9.10 embryonic hippocampal cells. Using ultrafast liquid chromatography tandem mass spectrometry, we showed that exosomes are rich in sphingomyelin species compared to whole cells. To interrogate the possible functions of vitamin D3, we established the optimal conditions of cell treatment and we analyzed exosome composition. Vitamin D3 was identified as responsible for the vitamin D receptor loss, for the increase in neutral sphingomyelinase content and sphingomyelin changes. As a consequence, the generation of ceramide upon vitamin D3 treatment was evident. Incubation of the cells with neutral sphingomyelinase, or the same concentration of ceramide produced in exosomes was necessary and sufficient to stimulate embryonic hippocampal cell differentiation, as vitamin D3. This is the first time that exosome ceramide is interrogated for mediate the effect of vitamin D3 in inducing cell differentiation.

## 1. Introduction

Exosomes are a subset of extracellular vesicles universally recognized as natural nanoparticles involved in numerous biologic processes and clinical diseases. They have specific proteins, lipids, RNA, and DNA compositions that can be transferred from a donor cell to a recipient allowing cell-to-cell communication useful for maintaining cell functions and tissue homeostasis [1]. Today, it is believed that exosomes originate by the succession of different biological moments: invagination of the plasma membrane, formation of intracellular multivesicular endosomes, fusion of these with the plasma membrane and release of the vesicular contents forming extracellular vesicles with diameter less than 150 nm or exosomes [2]. Lipids are the molecules mainly involved in the exosome formation [3]. Numerous cell types are capable of releasing exosomes in different biological fluids [4]. Once released, intercellular exosomes remodel the extracellular matrix, influence the trafficking of signal molecules [5] and enter neighboring cells. The content and composition of lipids, proteins, DNA and RNA (mRNA, and microRNA) directly reflect the metabolic state of the cells from which they originate [6]. mRNA and microRNA transferred via exosomes become functional in the new cell environment [4]. Thus, exosomes influence the fate of the recipient cells [1]. Due to their action, they have been considered the main players in different physiopathological conditions [4] and useful diagnostic or prognostic markers [7]. In cancer, exosomes are key mediators of the communication between malignant cells and their microenvironment. In chronic lymphocytic leukemia, the induction of myeloid-derived suppressor cells mediated by exosome miR-155 transfer and responsible for immune suppression is inhibited by vitamin D [8]. Recently, a mass spectrometric proteomics characterization of exosomes purified from the bronchoalveolar lavage fluid of patients with sarcoidosis and healthy control subjects was performed by identifying 690 proteins [9]. As expected, several inflammatory proteins were overexpressed. Interestingly, the study highlighted for the first time the abundance of vitamin D3 (VD3)-binding protein in sarcoidosis patients. Thus, exosome-associated VD3-binding protein was investigated in the plasma and the results indicated that it might be a useful biomarker for sarcoidosis. Moreover, exosome-associated VD3-binding protein from tear fluids was significantly increased in patients with thyroid eye disease, compared to controls, indicating that it might play a role in the disease pathogenesis [10].

Sphingolipids (SphLs) have long been considered only structural components of cell membranes but in time they have been appreciated as second messengers in a wide array of signaling pathways in cell proliferation, differentiation, senescence, apoptosis and, cancer [11]. The current body of work reveals that VD3 affects SphL metabolism and vice-versa [12]. The progenitor of the SphL family is the sphingomyelin (SM) that is degraded to ceramide (Cer) and phosphocholine by sphingomyelinase (SMase). Ceramide is catabolized by ceramidase (Cerase) to sphingosine (Sph) that is converted to sphingosine-1-phosphate (S1P) by sphingosine kinase (SphK). VD3 is able to regulate SMase and SphK, S1P and, S1P receptor [12]. The interactions between SphL levels and VD3 in the blood has been reported in children affected by asthma [13]. In the same paper, the induction of S1P by VD3 was obtained in human bronchial epithelial cells [13]. Moreover, VD3 supplementation in overweight/obese African American patients resulted in a high level of Cer serum [14]. In addition, VD3 treatment influences SphK, S1P and S1P receptor levels in primary monocytes of type 2 diabetes mellitus patients [15]. While not much is known about the exact mechanism of VD3-SphL crosstalk, there is evidence that in embryonic hippocampal cells (HN9.10e cell line) VD3 induces cell differentiation via VD3 receptor (VDR) located in inner nuclear membrane microdomains rich in SM and cholesterol content [16]. Here, VD3 links its receptor and regulates gene transcription [16]. Therefore, VD3 is able to induce HN9.10e cell differentiation, but its possible role in exosome release, which might be involved in this process, was not investigated.

The results of these few recent studies suggest that VD3: (1) might play an important role in exosome functions and (2) interplays with SphL pathway. Of interest, no studies have addressed the effect of VD3 in SphL composition of exosomes.

## 2. Results

### 2.1. Parameters for Cell and Exosome Measurements

In order to test whether physiological concentration of VD3 (100 nmol/L), previously used to induce HN9.10 cell differentiation [16,17] was the better concentration, we first set out to study the dose-dependent effect of VD3 on HN9.10 cell viability after 24 h of culture by using MTT assay. The results demonstrated that VD3 from 25 nmol/L to 300 nmol/L concentration did not induce damage in the cells, indicating that it is not a cytotoxic substance; 1% and 2% DMSO were used as positive controls. Only 400 nmol/L concentration was responsible for a low reduction of cell viability (Figure 1a).

Thus, after verifying the effect of the different concentrations, we have chosen to use the physiological concentration of VD3 (100 nmol/L), previously used in the same cells [16,17].

However, we wanted to confirm the effect of 100 nmol/L VD3 on cell viability. Trypan blue exclusion assay confirmed a good cell viability, with 6.8 + 1.7% of dead cells in CTR sample 8.6 + 1.6% in experimental cells (Figure 1a). The protein content was 382 ± 21 µg/10^6^ CTR cells with no significant changes after VD3 treatment. The exosome proteins were 8.11 ± 0.9% of cell proteins in CTR and 9.44 ± 0.2% in experimental sample (Figure 1b). Because CD9 and CD63 are exosome purification markers, we performed the western blot analysis of the two proteins by using the same protein amount for cells and exosomes. The results showed an undetectable amount of CD9 and CD63 in cells, indicating their very low content respect to total proteins. The high level of CD9 and CD63 in exosomes is index of a good purification (Figure 1c). The treatment with VD3 increases CD9 and even more CD63 expression (Figure 1c,d), suggesting a better purification under vitamin D3 treatment.

### 2.2. VD3 Causes VDR Suppression in Exosomes

VD3 is involved in many pathophysiological processes. However, its role in the exosomes is not defined yet. In the settings of embryonic hippocampal cell differentiation due to VD3, vitamin D receptor (VDR) was found abundantly expressed in inner nuclear membrane microdomains, which regulates gene expression [16]. Therefore, we questioned whether VDR was present in exosomes and whether VD3 was required for it. Thus, in order to investigate the presence of VDR in exosomes and to establish the effect of VD3, we performed an immunoblotting analysis of VDR in whole cells and exosomes. The VDR expression in HN9.10 cells was very low with respect to the total protein content and VD3 treatment did not induce a significant increase. Importantly, VDR was present in high concentration in exosomes, suggesting that exosomes might derive from a specific part of the cells in which VDR is localized. Treatment with VD3 caused a significant suppression in VDR content (Figure 2a,b).

### 2.3. Regulation of Exosome Sphingolipid Pathway by VD3

For many basic cellular processes, such as signaling, cell division, DNA replication, cell differentiation, VD3-nSMase interplays are critical [18]. Moreover, SM links CHO to form lipid membrane microdomains which act as a platform for VDR in the cell nucleus [16]. Therefore, we asked whether VD3 induced changes in the exosome SM content. Thus, we performed a specific sphingolipidomic analysis in untreated (control sample) and VD3-treated cells and exosomes.

We analyzed sphingolipid species, and phosphatidylcholine (PC) species for comparison, by using: 12:0 SM, 16:0 SM, 18:1 SM, 24:0 SM, sphingosine-1-phosphate (S1P), C18:0 sphinganine, C6:0 Cer, C8:0 Cer, C16:0 Cer, C18:0 Cer, C20:0 Cer, C24:0 Cer, C12:0 dihydroCer, arachidonoylglycerol (2AG), C16:0 glucosylceramide (GluCer), 16-0 18-1 phosphatidylcholine (PC), 16-0 20-4 PC, 18-1 18-0 PC external calibrators. The results showed that the 12:0 SM was present in traces and S1P, C18:0 sphinganine, C6:0 Cer, C8:0 Cer, C12:0 dihydroCer, 2AG were absent in exosomes. Therefore, we added to the samples 12:0 SM and 6:0 Cer as internal standards. The recovery of 12:0 SM was 83% and 85% in the control and VD3 cells, respectively, and 79% and 81% in the control and VD3 exosomes, respectively. The recovery of 6:0 Cer was 87% and 84% in the control and VD3 cells, respectively, and 83% and 71% in the control and VD3 exosomes, respectively. As shown in 16:0 SM, 18:1 SM were higher and 24:0 SM was lower in exosomes than in whole cells (Figure 3). To rule out a possible effect of VD3 on SM composition, we measured SM species after cell treatment with VD3 and purification of exosomes. VD3 induced a strong reduction of all species in both cells and exosomes.

Interestingly, ceramide was also higher in exosomes than in cells and VD3 induced a significant accumulation (Figure 4). The most represented species increased by about 200 ng/mg protein. Strikingly, exosomes showed the presence of glucosylceramide (GluCer) that was undetectable in whole cells (Figure 4).

Only to compare the behaviour of SM to phosphatidylcholine (PC) as another lipid containing phosphocholine group, the analysis of 16-0 18-1 PC, 16-0 20-4 PC, 18-1 18-0 PC was performed. The results highlighted the same content of 16-0 18-1 PC referred to protein in nuclei and exosomes and a high content 16-0 20-4 PC, 18-1 18-0 PC in exosomes. The VD3 treatment reduced PC species only in the exosomes (Figure 5). Therefore, data of PC and SM were not coincidences.

Since our data highlighted important changes, specifically in SM and Cer content and not in S1P and other analyzed sphingolipids, we focused the attention of nSMase expression and activity. Our data showed that nSMase/mg protein was more abundant in exosomes than in cells (Figure 6a) and its increase VD3-dependent was comparable in the two samples (Figure 6b). Next, the effects of VD3 on nSMase activity was assayed. The results revealed about a 1.3-fold increase in both cells and exosomes (Figure 6c) but whether the activity was referred to SM content, the increase was not evident (Figure 6d).

### 2.4. Effect of Ceramide on HN9.10 Cell Differentiation

It has been previously demonstrated that VD3 induced HN9.10 differentiation [17] Thus, the increase of Cer in exosomes upon VD3 treatment prompted us to explore the role of SMase and Cer in HN9.10 cell differentiation in order to establish whether the effect of VD3 was mediated by exosomes. We stably cultured the cells in the presence of SMase and Cer at the same concentration produced in the exosomes after VD3 treatment.

Biologically, SMase and Cer induced neurite formation highlighted by GFAP overexpression, similar to that obtained by VD3 (Figure 7).

## 3. Discussion

Classically, VD3 is known to have effect in the maintenance of calcium and phosphate homeostasis useful for the bone health. Over the course of the last decade, it has become increasingly clear that VD3 regulates multiple cellular processes as cell growth, differentiation, apoptosis and inflammation, having a plethora of pleiotropic effects in normal and malignant cells [20]. Thus, VD3 is implicated in diabetes mellitus, kidney, cardiovascular and immune diseases [20], in allergy [21] and, in neurodegenerative disorders [22]. We have previously demonstrated that VD3 induces HN9.10 differentiation [19] via VD3-VDR interaction in nuclear lipid microdomains rich in cholesterol and SM content [20]. Given the interest in exosomes in cell-to-cell communication, it is perhaps surprising that the role of VD3 in exosome lipid composition has not been investigated. Here, we addressed this knowledge gap using HN9.10 cells to investigate whether their differentiation VD3-induced is related or not with the change of exosome lipid composition. Biologically, the results show that exosomes released from HN9.10 are composed by a high amount of SM and Cer. VD3 increases nSMase content in exosomes with consequent reduction in SM species and production of Cer species. Functionally, this translates into an impaired cell growth ability and into cell differentiation showed by GFAP immunostaining. In HN9.10 cells, the enhancement of differentiation was associated with both VD3 and Cer treatment, suggesting that the action of VD3 might be via the enrichment of exosome Cer. It has been widely demonstrated that Cer was involved in cell differentiation [22,23].

Interestingly, the exosomes contained a high level of VDR and GluCer with respect to whole cells. This data could be of great interest in the future for several reasons: (1) VDR and GluCer could be important for cell-to-cell communication; (2) VDR could be a marker for studying the biogenesis of exosomes. Importantly, treating HN9.10 cells with VD3, to our surprise, induced a loss of VDR and GluCer in exosomes. Overall, this functionally links exosome lipid remodelling and suggests that VD3 would be effective in inducing cell differentiation by changing the structure of the exosome. Thus, our major results demonstrated a role for VD3 in exosome lipid composition, showing that VD3 led to both SM decrease, which could be attributed the loss of VDR and GluCer, and consequent Cer accumulation, which might be responsible for cell differentiation. These findings agree with prior studies reporting the effect of VD3 in HN9.10 cell differentiation, and the capacity of exogenous Cer to induce cell differentiation [19,22,23,24]. Thus, here we propose to connect the release of exosomes with a high content of Cer upon VD3 treatments to the effect of VD3 on cell differentiation. Therefore, the sphingolipid composition of exosomes might be relevant in cell–cell communication by favoring the differentiation of nearby cells. The possible significance of the VDR and GluCer loss is a significant question that is currently being explored.

## 4. Materials and Methods

### 4.1. Reagents

Dulbecco’s modified Eagle’s medium (DMEM), bovine serum albumin, dithiothreitol, phenylmethylsulfonylfluoride, nSMase were obtained from Sigma Chemical, Co. (St. Louis, MO, USA); C6-Cer was obtained by Avanti Polar (Alabaster, AL, USA); VD3 was obtained from DBA Italia (Segrate, Milan, Italy); anti-glial fibrillary acid protein (GFAP) antibody was obtained from Dako, Agilent (Santa Clara, CA, USA); anti-VDR and anti-SMase from Elabscience (Houston, TX, USA); anti-CD9 and anti-CD63 from Biorbyt (Cambridge, UK). For research involving biohazards, biological select agents and reagents, standard biosecurity safety procedures were carried out.

### 4.2. Cell Culture and Treatments

Immortalized hippocampal neurons HN9.10e (kind gift of Kieran Breen, Ninewells Hospital, Dundee, UK) were grown in DMEM supplemented with 10% FBS, 2 mM l-glutamine, 100 IU/mL penicillin, 100 μg/mL streptomycin, and 2.5 μg/mL amphotericin B [16]. Cells were maintained at 37 °C in a saturating humidity atmosphere containing 95% air and 5% CO_2_. To study of the VD3 effect on exosomes, VD3 dissolved in absolute ethanol as vehicle at the 100 nmol/L physiological concentration, was added to the cultures for 24 h; in control samples only absolute ethanol was added [19]. To study the effect of neutral sphingomyelinase (N-SMase) or ceramide (Cer) on cell differentiation, cells were cultured in the presence of 6U SMase/mg prot or 200 ng Cer /mg protein.

### 4.3. Cell Viability

MTT assay was used to test cellular viability, as previously reported [19]. HN9.10 cells were seeded into 96-well plates (1 × 10^4^ cells/well density) with DMEM complete medium. After 24 h, the culture medium was removed and fresh complete medium containing VD3 at different concentrations (25, 50, 75, 200, 300, and 400 nmol/L) was added; the cells were incubated for 24 h. Then, MTT reagent was dissolved in PBS 1× and added to the culture at 0.5 mg/mL final concentration. After 3 h incubation at 37 °C, the supernatant was carefully removed, and formazan salt crystals were dissolved in 200 µL DMSO that was added to each well. The absorbance (OD) values were measured spectrophotometrically at 540 nm using an automatic microplate reader (Eliza MAT 2000, DRG Instruments, GmbH, Marburg, Germany). Each experiment was performed twice in triplicate. Cell viability was expressed as a percentage relative to the control cells.

### 4.4. Trypan Blue Exclusion Assay

Based on the MTT results, the 100 nmol/L dose was chosen and a cell count by the trypan blue exclusion assay was performed using a Countess™ (Invitrogen Srl, Milan, Italy) automated cell counter, as previously reported [19]. Briefly, 50 μL of VD3-treated NH9.10 cell suspension (5 × 10^4^/500 μL) was mixed with equal volumes of 0.4% trypan blue and loaded onto a Countess cell-counting chamber slide. Untreated cells were used as control. Images were captured by a camera and then analyzed with image analysis software to automatically measure cell count and viability.

### 4.5. Exosome Isolation

Exosomes released in the culture medium from untreated HN9.10 cells (control) and VD3-treated HN9.10 cells were prepared by “total exosome isolation” kit from Invitrogen-Thermo Fisher Scientific (Waltham, MA, USA) by following the instructions of the company

### 4.6. Protein Content

Total protein concentration was determined spectrophotometrically at 750 nm by using bovine BSA as a standard, as previously reported [19].

### 4.7. Electrophoresis and Western Blot Analysis

The analysis of protein expression was performed as previously reported [19]. Briefly, 60 micrograms of protein were loaded on SDS–PAGE using 10% running gel. The transfer of protein was carried out onto nitrocellulose in 90 min [17]. The membranes were blocked for 30 min with 5% non-fat dry milk in PBS (pH 7.5) and incubated overnight at 4 °C with specific antibodies. The blots were treated with horseradish-conjugated secondary antibodies for 90 min. Band detection was performed using enhanced chemiluminescence kit from Amersham Pharmacia Biotech (Rainham, Essex, UK). A densitometric analysis was performed by Chemidoc Imagequant LAS500–Ge Healthcare-Life Science (Milano, Italy).

### 4.8. Sphingomyelinase Activity Assay

The SMase activity was assayed as previously reported [25]. Cells were suspended in 0.1% NP-40 detergent in PBS, sonicated for 30 s on ice at 20 watt, kept on ice for 30 min and centrifuged at 16,000 *g* for 10 min. The supernatants were used for aSMase assay. Then, 60 μg/10 μL proteins were incubated with 10 μL HMU-PC substrate for 10 min at 37 °C. The reaction was stopped by adding 200 μL stop buffer. The fluorescence of 6-hexadecanoyl-4-methylumbelliferone (HMU) was measured with FLUOstar Optima fluorimeter (BMG Labtech, Germany), by using the filter set of 4-methylumbelliferone (MU), 360 nm excitation, and 460 nm emission. The fluorimeter was calibrated with MU in stop buffer.

### 4.9. Ultrafast Liquid Chromatography–Tandem Mass Spectrometry

Lipid extraction and analysis was performed as previously reported [26,27]. The pellets of the cells were suspended in Tris 10 mM, pH 7.4, and diluted with 1 mL methanol. Three milliliters of ultra-pure water and 3 mL methyl tert-butyl ether (MTBE) were added. Each sample was vortexed for 1 min and centrifuged at 3000× *g* for 5 min according to Matyash et al. [28]. The supernatant was recovered. The extraction with MTBE was repeated on the pellet and the supernatant was added to the first. The organic phase was dried under nitrogen flow and resuspended in 500 μL of methanol. The: 12:0 SM, 16:0 SM, 18:1 SM, 24:0 SM, sphingosine-1-phosphate, C18:0 sphinganine, C6:0 Cer, C8:0 Cer, C16:0 Cer, C18:0 Cer, C20:0 Cer, C24:0 Cer. C12:0 dihydroCer, arachidonoylglycerol (2AG), C16:0 glucosylceramide (GluCer), 16-0 18-1 phosphatidylcholine (PC), 16-0 20-4 PC, 18-1 18-0 PC standards were dissolved in chloroform/methanol (9:1 vol/vol) at 10 μg/mL final concentration. The stock solutions were stored at −20 °C. Working calibrators were prepared by diluting stock solutions with methanol to 500:0, 250:0, 100:0, and 50:0 ng/mL final concentrations. Twenty microliters of external standards or lipids extracted from serum with 12:0 SM and 6:0 Cer as internal standards (500 ng/mL) were injected after purification with specific nylon filters (0.2 μm). The analyses were carried out by using the Ultra Performance Liquid Chromatography system tandem mass spectrometer (Applied Biosystems, Italy). The lipid species were separated, identified, and analyzed as previously reported [27,29] according to Rabagny et al. [30]. Liquid Chromatography system was Shimadzu Prominence UFLC, the pump was Shimadzu LC-20 AD, the detector was API 3200 linear triple quadrupole MS/MS, the injection valve was Valco valve, the autosampler was Schimadzu SR-20 AC HT, the column temperature stabilizer was Schimadzu CTO-20A. The samples were separated on a Phenomenex Kinetex phenyl-hexyl 100 A column (50 × 4.60-mm diameter, 2.6-μm particle diameter) with a precolumn security guard Phenomenex ULTRA phenyl-hexyl 4.6. Column temperature was set at 50 °C and the flow rate at 0.9 mL/min. Solvent A was 1% formic acid; solvent B was 100% isopropanol containing 0.1% formic acid. The run was performed for 3 min in 50% solvent B and then in a gradient to reach 100% solvent B in 5 min. The system needed to be reconditioned for 5 min with 50% solvent B before the next injection. The lipid species were identified by using positive turbo-ion spray (ESI) and modality multipole-reaction monitoring. Ion spray voltage was 5.4 kV, gas 1 was air, gas 2 was nitrogen, temperature was 650 °C, and the flow rate curtain gas was 40.5 L/h. The flow of the collision gas was maintained at 5.0 L/h. Data were acquired and processed using AnalystTM and Analyst 1.5 software in a Dell Precision T3400 personal computer with a Samsung ML-2851 MD graphical printer. All parameters of the analyzed molecules were reported in the Figure 8a. The lipid species under study were reported in Figure 8b.

### 4.10. Immunofluorescence

Cells cultured in the presence of VD3 or nSMase or C6-Cer were treated and incubated as previously reported [31]. Briefly, cells were incubated with anti-GFAP primary antibodies diluted 1:100 in 3% (*w/v*) BSA in PBS for 1 h, washed three times in 0.1% (*v/v*) Tween-20 in PBS and twice in PBS, incubated with tetramethylrhodamine isothiocyanate (TRITC)-conjugated anti-rabbit IgG for 1 h, diluted 1:50 in 3% (*w/v*) BSA in PBS and washed as above. The diamidino-2-phenylindole (DAPI) nuclear counterstain was used. The samples were mounted in 80% (*w/v*) glycerol, containing 0.02% (*w/v*) NaN_3_ and *p*-phenylenediamine (1 mg/mL) in PBS to prevent fluorescence fading. The antibody incubations were done in a humid chamber at room temperature. Fluorescent analysis was performed on a DMRB Leika epi-fluorescent microscope equipped with a digital camera. The intensity of immunofluorescence was evaluated with Scion Image.

### 4.11. Statistical Analysis

Data were expressed as means ± SD of three independent experiments and their significance was checked by Student’s *t*-test to analyze experimental samples versus control sample and ANOVA test to analyze exosomes versus cells in experimental samples versus control sample.

## Figures and Tables

**Figure 1 ijms-22-09287-f001:**
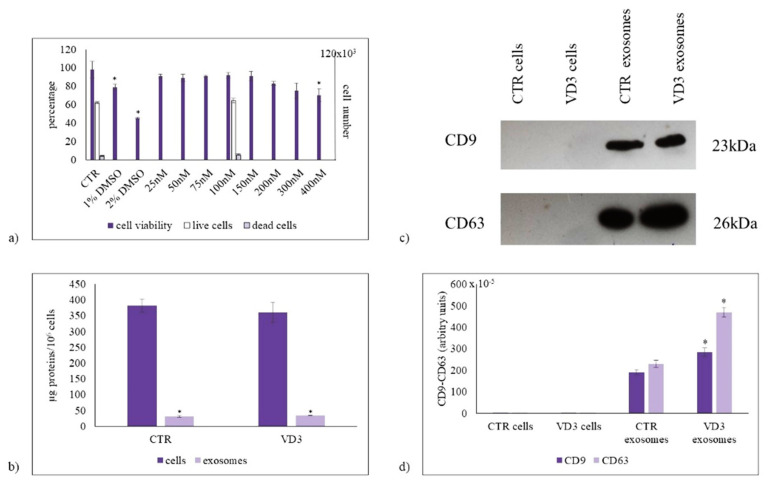
The effect of vitamin D3 on cell viability and exosome purification. (**a**) Left ordinate, effect of VD3 on HN9.10 cell viability (in gray). Cells were cultured with increasing doses of VD3 from 25 nmol/L to 400 nmol/L for 24 h and the viability was measured by MTT assay. Values were reported as percentage viability of the control sample (CTR). 1% DMSO and 2% DMSO were used as positive controls; right ordinate, live cells (in white) and dead cells (in light violet) under 100 nmol/L VD3 treatment evaluated by trypan blue exclusion assay; (**b**) Protein content in cells and exosomes; (**c**) Immunoblotting analysis of CD9 and CD63, as markers of exosome purification. The position of the 23 kDa for CD9, and 26 kDa for CD63 was evaluated in relation to the molecular-weight size markers. (**d**) The area density was quantified by Chemidoc Imagequant LAS500 by specific IQ programm. (case number:6) CTR, control sample; VD3, vitamin D3 sample. Data were expressed as mean ± SD of three independent experiments performed in duplicate. * *p* < 0.05 versus the control sample.

**Figure 2 ijms-22-09287-f002:**
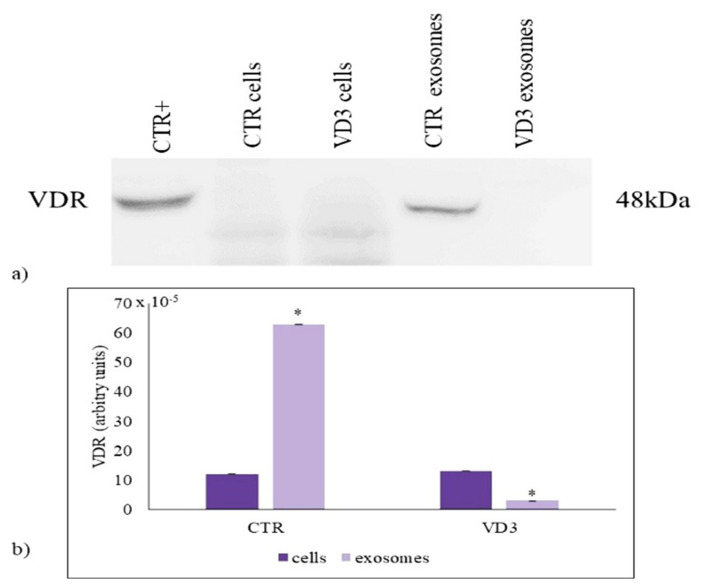
VDR content in cells and exosomes. (**a**) Immunoblotting analysis. The position of the 48 kDa was evaluated in relation to the molecular-weight size markers. NCI-N87 cells were used as positive control [17]; (**b**) The area density was quantified by Chemidoc Imagequant LAS500 by specific IQ programm. Data were expressed as mean ± SD of three independent experiments performed in duplicate (case number: 6). * *p* < 0.05 versus the control sample. CTR, control sample; VD3, vitamin D3 sample.

**Figure 3 ijms-22-09287-f003:**
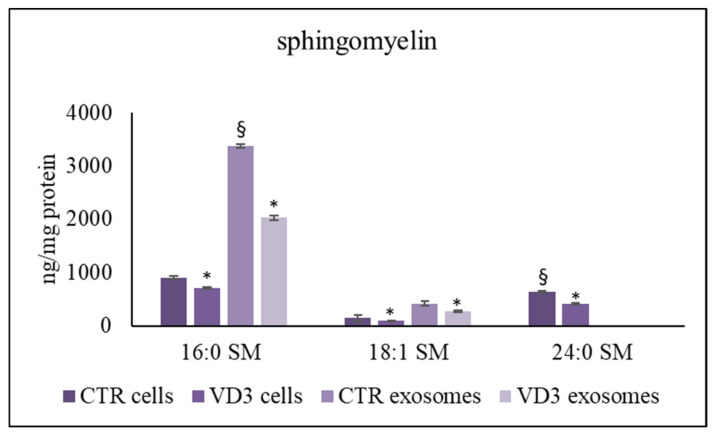
UFLC MS/MS analysis of SM species in cells and exosomes. The analysis was performed as reported in the Section 4. Data are expressed as ng/mg protein and represent the mean ± SD of three independent experiments performed in duplicate (case number: 6). * *p* < 0.05 versus the control sample, ^§^
*p* < 0.05 versus cells CTR, control sample; VD3, vitamin D3 sample.

**Figure 4 ijms-22-09287-f004:**
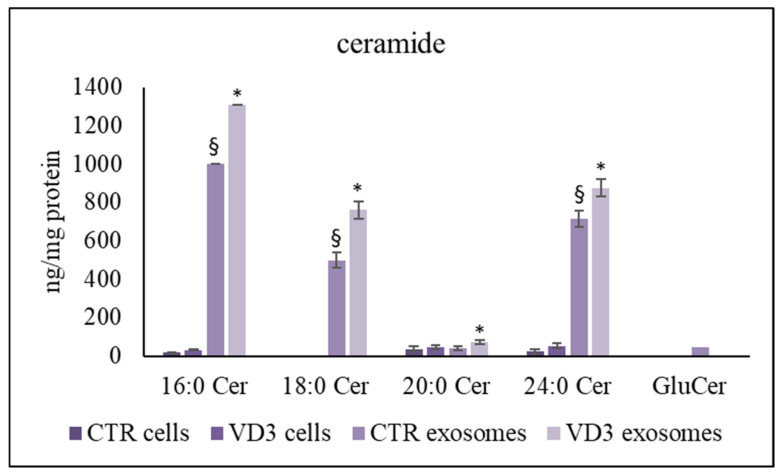
UFLC MS/MS analysis of ceramide species in cells and exosomes. The analysis was performed as reported in the Section 4. Data are expressed as nmol/mg protein. Data represent the mean ± SD of three independent experiments performed in duplicate (case number: 6). * *p* < 0.05 versus the control sample, ^§^
*p* < 0.05 versus cells Cer, ceramide; GluCer, glucosylceramide; CTR, control sample; VD3, vitamin D3 sample.

**Figure 5 ijms-22-09287-f005:**
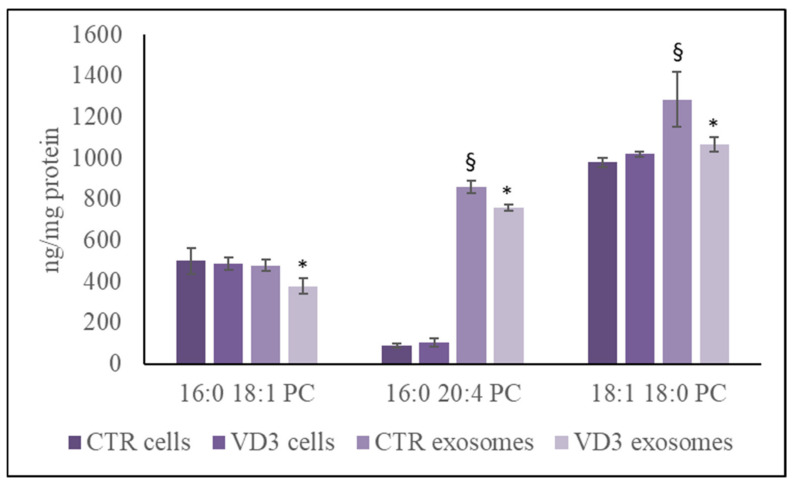
UFLC MS/MS analysis of phosphatidylcholine species in cells and exosomes. The analysis was performed as reported in the Section 4. Data are expressed as nmol/mg protein. Data represent the mean ± SD of three independent experiments performed in duplicate (case number: 6). * *p* < 0.05 versus the control sample, ^§^
*p* < 0.05 versus cells PC, phosphatidylcholine; CTR, control sample; VD3, vitamin D3 sample.

**Figure 6 ijms-22-09287-f006:**
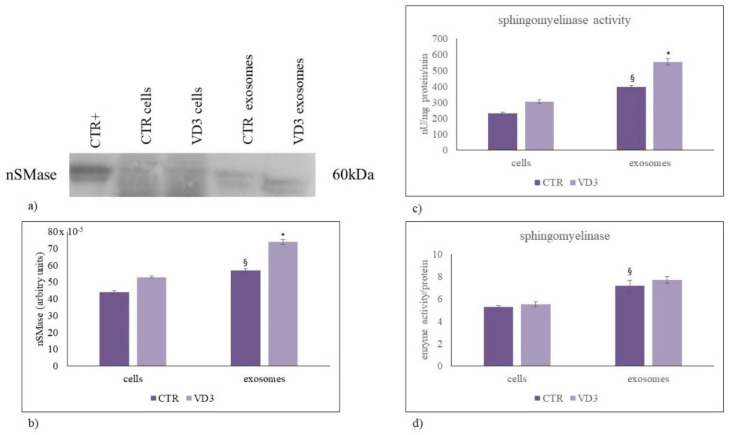
Neutral sphingomyelinase in cells and exosomes. (**a**) Immunoblotting analysis. The position of the 60 kDa was evaluated in relation to the molecular-weight size markers. HaCaT cells were used as positive control [19]; (**b**) The area density was quantified by Chemidoc Imagequant LAS500 by specific IQ programm. (**c**) Enzymatic activity of neutral sphingomyelinase. Data are expressed as nU/mg protein/min; (**d**) Neutral sphingomyelinase activity in relation to enzyme content. Data represent the mean ± SD of three independent experiments performed in duplicate (case number: 6). * *p* < 0.05 versus the control sample, ^§^
*p* < 0.05 versus cells, CTR, control sample; VD3, vitamin D3 sample.

**Figure 7 ijms-22-09287-f007:**
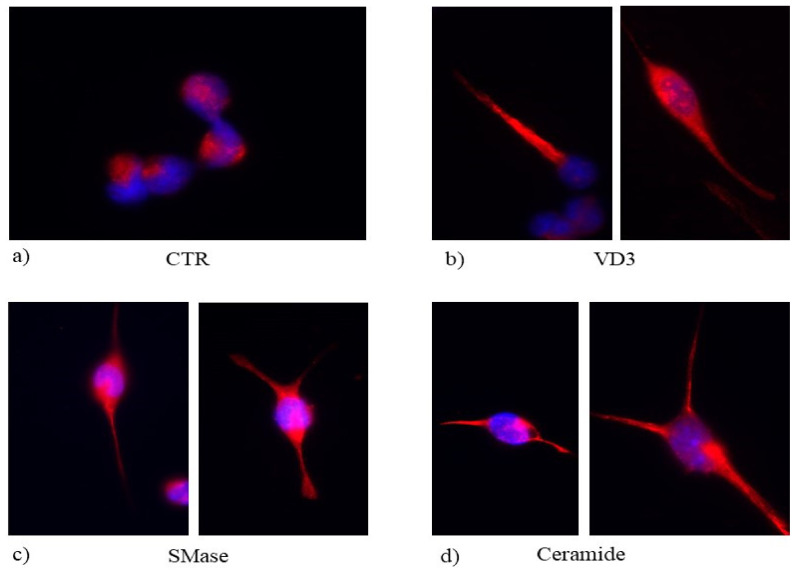
Immunofluorescence of H9.10e cells without (CTR, (**a**) or with VD3 (**b**), SMase (**c**) and ceramide (**d**). The image represents the merged signal of GFAP immunolabelling, in red, counterstained with DAPI (in blue). 100× magnification.

**Figure 8 ijms-22-09287-f008:**
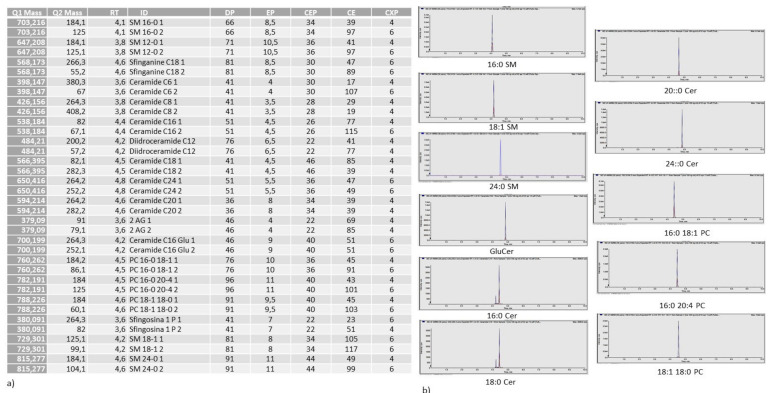
(**a**) Table showing lipidomic data analysis. main transitions decimal (Q1, parent ion; Q2 daughter ions. Retention times (RT); molecule name (ID); declustering potential (DP); entrance potential (EP); collision cell entrance potential (CEP); collision energies (CE); collision cell exit potential (CXP). RT (min); DP, EP, CEP, CE and CXP (V). (**b**) molecules under study.

## Data Availability

Not applicable.

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
