# Peer review of "Vitamin D3 Enriches Ceramide Content in Exosomes Released by Embryonic Hippocampal Cells"

_ijms, 2021, doi:10.3390/ijms22179287_

Round 1

Reviewer 1 Report

The current version of the paper allows for its publication.

Author Response

Thank you very much for your help

Reviewer 2 Report

The paper was not well written, the study was not well organized, and the figure quality was not good. Each of the experiments was only triplicated and lacked controls. The study design lacked controls or inhibitors. Statistics were inappropriate. A reader is hard to catch the clinical significance and implication of this article. Other concerns are listed below:

  1. In Figure 1, Seeding 1 × 104 cells/well density, after 24 hours, the cell number was 120 × 103 cells? Why only calculate live cells and dead cells for control and 100 nmol/L, but not for other treatment doses? The quality of the presentation was awful, readers can't catch the authors' ideas.
  2. In Figure 1, what was the n value? The authors need to indicate the numbers of each group and for each figure. Besides, a student's t-test can not be used for Nonparametric statistics.
  3. Why the expression levels of CD9 and CD63 were elevated by treating with Vitamin D3? What were the controls of exosomes? Were the isolated exosomes neural-derived? 
  4. In Figure 2, what was the internal control of cells and exosomes? I can't see a clear VDR band in CTR cells. What is the statistic method for an n=3 experiment?
  5. In Figure 3, what is the molecular weight of nSMase? What was the positive control of nSMase? What was the internal control?
  6. How about sphingosine and sphingosine 1-phosphate? Was cell growth because of ceramide or sphingosine 1-phosphate? Why detect nSMase only? Why the authors not detect aSMase, sphingosine kinase...etc?

Author Response

Thank you very much for your observations that allow me to clarify some points along the test and that significantly improved the manuscript.

The paper was not well written, the study was not well organized, and the figure quality was not good. Each of the experiments was only triplicated and lacked controls. The study design lacked controls or inhibitors. Statistics were inappropriate. A reader is hard to catch the clinical significance and implication of this article.

The paper show results on the effect of vitamin D3 on sphingolipid composition of exosomes. Therefore the paper has been organized in the following parts: 1) Introduction: exosomes, sphingolipids and interplay sphingolipid/vitamin D3; 2) Results: effects of vitamin D3 on cells and exoxomes; effects of vitamin D3 on vitamin D3 receptor in cells and exoxomes;  effect of vitamin D3 on sphingolipid pathway in exosomes;  since ceramide resulted the most sphingolipid changed with vitamin D3 treatment, it was studied the effect of ceramide and sphingomyelinase on cell differentiation. The experimental model was clarified, by including specific statements in each paragraph of the result section and by reorganizing the order of the paragraphs (lines 93,94; 144-146; 165,166; 226-228; 245-247).

When necessary, the figures have been enlarged to increase the quality (figures 2 and 5).

We have performed three independent experiments in duplicate (n 6) since the results were strongly repetitive; it is not a clinical study with many variabilities. Controls were untreated cells, as reported in all results and figures. Statistical analysis was performed by t-test comparing experimental samples versus control samples or exosomes versus cells. In the present version, also ANOVA test was used (p14 lines 14-17).

The article suggests that the effect of vitamin D3 on embryonic hippocampal cell differentiation, already demonstrated in other papers, is mediated by the change of sphingolipid composition of exosomes that act in cell-cell communication. It has been clarified (lines 286-289).

Other concerns are listed below:

  1. In Figure 1, Seeding 1 × 104 cells/well density, after 24 hours, the cell number was 120 × 103cells?

I’m sorry for not clear legend. In the ordinate on the right, the number of live and death cells was reported. The test used for live and dead cells was “Trypan Blue Exclusion Assay”. As reported in the method NH9.10 cells were 5 × 104/500 μL or 50 x 103. Thus, in the right ordinate 120x103cells was reported. Considering the cell growth, after 24 h of culture the live cells about 60 x 103. The legend has been clarified (lines 117, 120)

  1. Why only calculate live cells and dead cells for control and 100 nmol/L, but not for other treatment doses? The quality of the presentation was awful, readers can't catch the authors' ideas.

Thank you for this observation. 100 nmol/L was the physiological dose that was used to induce embryonic hippocampal cell differentiation in previous articles. Therefore, other concentrations were used to test whether they had a different effect on cell vitality but since the results were similar, 100 nmol/L concentration was used in all experiments. This point was clarified (lines 93, 94, 103)

  1. In Figure 1, what was the n value? The authors need to indicate the numbers of each group and for each figure. Besides, a student's t-test can not be used for Nonparametric statistics.

Three independent experiments performed in duplicate was performed, as reported in the figure legend, therefore n value is six. The statistic method has been reported in materials and methods. All legends were rewritten

  1. Why the expression levels of CD9 and CD63 were elevated by treating with Vitamin D3?

Since CD9 and CD63 were markers of exosome purification, it is possible to suppose that vitamin D3 permits a better purification. It has been included (lines 113, 114)

What were the controls of exosomes?

Exosomes were purified from cell culture medium of untreated (control) or vitamin D3 treated cells. It has been highlighted (lines 334,335)

Were the isolated exosomes neural-derived?

Yes, they were released from the HN9.10 cells (Lines 334,335)

  1. In Figure 2, what was the internal control of cells and exosomes? I can't see a clear VDR band in CTR cells. What is the statistic method for an n=3 experiment?

As control of cells that had VDR, NCI-N87 cells was used. By using the same protein amount of cells and exosomes, we demonstrated that the VDR was low expressed in cells, respect to the total protein (lines 146-148). It was particularly concentrated in exosomes, by suggesting that exosomes derive from a specific part of the cells where VDR is localized. It has been included (lines 149-150). The statistic method has been reported in materials and methods

  1. In Figure 3, what is the molecular weight of nSMase? What was the positive control of nSMase? What was the internal control?

The molecular weight 60kDa, as reported in Figure  3a on the right (Fig.6a in the present version). HaCaT  cells were used as positive control, as reported in the legend

  1. How about sphingosine and sphingosine 1-phosphate? Was cell growth because of ceramide or sphingosine 1-phosphate?

As reported in the results, sphingosine 1-phosphate was absent in exosomes. Only a high level of ceramide and not sphingosine 1-phosphate was found in exosomes released from the cells treated with vitamin D3

As reported in the test, we analysed 12:0 SM,  16:0 SM, 18:1 SM, 24:0 SM, sphingosine-1-phosphate (S1P), C18:0 sphinganine, C6:0 Cer, C8:0 Cer, C16:0 Cer, C18:0 Cer, C20:0 Cer, C24:0 Cer, C12:0 dihydroCer,  arachidonoylglycerol (2AG), C16:0 glucosylceramide (GluCer), 16-0 18-1 phosphatidylcholine (PC), 16-0 20-4 PC, 18-1 18-0 PC.

Why detect nSMase only? Why the authors not detect aSMase, sphingosine kinase...etc?

We detected only nSMase and not sphingosine kinase because of only ceramide was increased in exosomes after vitamin D3 treatment of the cells. We did not analyzed aSMase because of exosomes work with neutral pH. Thank you for this observation! To better clarify this point, the analysis of nSMase was moved under sphingolipidomic analysis (fig.3 is now fig. 6).

This manuscript is a resubmission of an earlier submission. The following is a list of the peer review reports and author responses from that submission.

Round 1

Reviewer 1 Report

The manuscript by Carmela Conte et al. described a correlation between exosome ceramide and vitamin D3. 

The paper is based on rich literature (22 items, 86% from the last ten years).

Fig. 4b and Fig. 4d are too small to exact review. 

Lack of explaining some symbol using in Figure 5. 

Under Fig. 4

Instead (Under Fig. 4):

 c) Total saturated and unsaturated SM species. Data are expressed as μ g/mg protein; d) effect of Vd3 on SM species

Should be (Under Fig. 4):

c) Total saturated and unsaturated SM species. Data are expressed as μg/mg protein; d) Effect of VD3 on SM species.

Instead:

nM

Should be:

nmol/L

Instead (in Body Text):

Effect of VD3 on HN9.10 cell viability (in gray). a) Cells were cultured with increasing doses of VD3 from 25 nM to 400 nM for 24 h and the viability was measured by MTT assay. Values were reported as percentage viability of the control sample (CTR). 1% DMSO and 2% DMSO were used as positive controls. Live cells (in white) and dead cells (in light violet) evaluated by trypan blue exclusion assay; b) protein content in cells and exosomes; c) Immunoblotting analysis of CD9 and CD63, as markers of exosomes. The position of the 23 kDa for CD9, and 26 kDa for CD63 was evaluated in relation to the molecular-weight size markers. d)The area density was quantified by Chemidoc Imagequant LAS500 by specific IQ programm. Data were expressed as mean ± SD of three independent 118experiments performed in duplicate. *p< 0.05 versus the control sample

Should be (under Figure 1):

Effect of VD3 on HN9.10 cell viability (in gray). a) Cells were cultured with increasing doses of VD3 from 25 nmol/L to 400 nmol/L for 24h, and the viability was measured by MTT assay. Values were reported as percentage viability of the control sample (CTR). 1% DMSO and 2% DMSO were used as positive controls. Live cells (in white) and dead cells (in light violet) evaluated by trypan blue exclusion assay; b) protein content in cells and exosomes; c) Immunoblotting analysis of CD9 and CD63, as markers of exosomes. The position of the 23 kDa for CD9, and 26 kDa for CD63 was evaluated in relation to the molecular-weight size markers. d)The area density was quantified by Chemidoc Imagequant LAS500 by a specific IQ programme. Data were expressed as mean ± SD of three independent 118experiments performed in duplicate. *p< 0.05 versus the control sample.

Furthermore,

Instead:

Albi et l., 2018

Should be:

Albi et al., 2018

Instead:

Patria et l., 2019

Should be:

Patria et al., 2019

I would also advise carefully revise English phrasing. It will improve the quality of the manuscript (For example, there are such phrases in body text:

(…)Since VD3 treatment induced HN9.10 cell differentiation and reduced SM content by producing 200ng Cer/mg exosomal protein of exosome released in the medium from 25x106cells, we decided to explore the role of SMase and Cer inHN9.10 cell differentiation 190in order to establish whether the effect of VD3 is mediated by exosomes.

(…) Briefly, sixty micrograms of protein were loaded on SDS–PAGE using10% running gel.The transfer of protein was carried out onto nitrocellulose in 90 min according.

Author Response

The manuscript by Carmela Conte et al. described a correlation between exosome ceramide and vitamin D3. The paper is based on rich literature (22 items, 86% from the last ten years).

Fig. 4b and Fig. 4d are too small to exact review. 

Big figures 4b and 4d have been included

Lack of explaining some symbol using in Figure 5

Explanation of symbols have been included

Under Fig. 4

Instead (Under Fig. 4):

  1. c) Total saturated and unsaturated SM species. Data are expressed as μ g/mg protein; d) effect of Vd3 on SM species.

Should be (Under Fig. 4):

  1. c) Total saturated and unsaturated SM species. Data are expressed as μg/mg protein; d) Effect of VD3 on SM species.

 It has been corrected

Instead: nM Should be: nmol/L.

It has been corrected along the text

Instead (in Body Text):

Effect of VD3 on HN9.10 cell viability (in gray). a) Cells were cultured with increasing doses of VD3 from 25 nM to 400 nM for 24 h and the viability was measured by MTT assay. Values were reported as percentage viability of the control sample (CTR). 1% DMSO and 2% DMSO were used as positive controls. Live cells (in white) and dead cells (in light violet) evaluated by trypan blue exclusion assay; b) protein content in cells and exosomes; c) Immunoblotting analysis of CD9 and CD63, as markers of exosomes. The position of the 23 kDa for CD9, and 26 kDa for CD63 was evaluated in relation to the molecular-weight size markers. d)The area density was quantified by Chemidoc Imagequant LAS500 by specific IQ programm. Data were expressed as mean ± SD of three independent 118experiments performed in duplicate. *p< 0.05 versus the control sample

Should be (under Figure 1):

Effect of VD3 on HN9.10 cell viability (in gray). a) Cells were cultured with increasing doses of VD3 from 25 nmol/L to 400 nmol/L for 24h, and the viability was measured by MTT assay. Values were reported as percentage viability of the control sample (CTR). 1% DMSO and 2% DMSO were used as positive controls. Live cells (in white) and dead cells (in light violet) evaluated by trypan blue exclusion assay; b) protein content in cells and exosomes; c) Immunoblotting analysis of CD9 and CD63, as markers of exosomes. The position of the 23 kDa for CD9, and 26 kDa for CD63 was evaluated in relation to the molecular-weight size markers. d)The area density was quantified by Chemidoc Imagequant LAS500 by a specific IQ programme. Data were expressed as mean ± SD of three independent 118experiments performed in duplicate. *p< 0.05 versus the control sample.

nmol/L has been corrected (line 91 and following)

Furthermore,

Instead:

Albi et l., 2018

 Should be:

Albi et al., 2018

It has been corrected (line 132)

Instead:

Patria et l., 2019

Should be:

Patria et al., 2019

It has been corrected (line 147)

I would also advise carefully revise English phrasing. It will improve the quality of the manuscript (For example, there are such phrases in body text:

(…)Since VD3 treatment induced HN9.10 cell differentiation and reduced SM content by producing 200ng Cer/mg exosomal protein of exosome released in the medium from 25x106cells, we decided to explore the role of SMase and Cer inHN9.10 cell differentiation 190in order to establish whether the effect of VD3 is mediated by exosomes.

It has been corrected (lines 213-217)

(…) Briefly, sixty micrograms of protein were loaded on SDS–PAGE using10% running gel.The transfer of protein was carried out onto nitrocellulose in 90 min according.

It has been corrected (line 311)

The English language has been revised

Reviewer 2 Report

Conte and co-workers examined the sphingolipid composition of exosomes released from hippocampus-derived cells in response to stimulation with vitamin D3.

In its current state, I do not consider the manuscript suitable for publication in a journal with the quality and reach of IJMS.

There are several reasons for this. First, the general appearance of the manuscript needs substantial improvement. The introduction is a mere compilation of findings from the literature, some of which are not related to the current paper. Why, for example, hippocampal cells are examined is not clear from the introduction. Furthermore, the quality of the figures is not good, some data are difficult to recognize and therefore to evaluate even after enlargement in the PDF. The list of references is incomplete, i.e. more references were cited in the text than are in the list.

The last point brings me to my biggest criticism, the sphingolipid analysis by LC-MS/MS. Crucial information is missing in the methods section. Two papers (presumably from the same working group) are cited, but they are not listed and thus could not be considered in the review process. The description of the methodology causes me to doubt the quality of the MS data and thus the foundation of this study.

The authors do not mention any internal standard substances – e.g. stable isotope-labeled SM and Cer analogues - which they add to the lipid extraction. However, this is absolutely common practice in mass spectrometric lipidomics studies. While the HPLC settings are described quite extensively, important information on the identification and quantification of SM and Cer are missing. What mass transitions were analyzed for the sphingomyelin and ceramide subspecies? What reference substances were used to confirm the identity of the subspecies? Was a saponification step included during lipid extraction to avoid interference with phosphatidylcholines (PCs)? This is quite crucial since SMs and PCs generate analogous fragment ions by CID, e.g. m/z 184 and 86. Can the authors safely rule out the possibility that overlap of SM and PC isotope signals occurred? For example, precursor ions of C20:0 SM (m/z 759.6) and PC 34:1 (m/z 760.6) differ by only 1 Da. In case of resubmission, authors should answer these open questions in sphingolipid quantification to make the compiled data appear more credible.

Author Response

Conte and co-workers examined the sphingolipid composition of exosomes released from hippocampus-derived cells in response to stimulation with vitamin D3. In its current state, I do not consider the manuscript suitable for publication in a journal with the quality and reach of IJMS.

There are several reasons for this. First, the general appearance of the manuscript needs substantial improvement. The introduction is a mere compilation of findings from the literature, some of which are not related to the current paper.

You are right but we introduced papers on exosomes in pathophysiology to highlight their role in cell fate

Why, for example, hippocampal cells are examined is not clear from the introduction.

It has been clarified (lines 85-89)

Furthermore, the quality of the figures is not good, some data are difficult to recognize and therefore to evaluate even after enlargement in the PDF.

the quality of difficult-to-interpret figures has been improved (see figures 4b and 4d)

The list of references is incomplete, i.e. more references were cited in the text than are in the list.

I'm really very sorry, it was our mistake. We have included an old bibliographic list in the final version. I apologize again. Now the list is complete

The last point brings me to my biggest criticism, the sphingolipid analysis by LC-MS/MS. Crucial information is missing in the methods section. Two papers (presumably from the same working group) are cited, but they are not listed and thus could not be considered in the review process. The description of the methodology causes me to doubt the quality of the MS data and thus the foundation of this study. The authors do not mention any internal standard substances – e.g. stable isotope-labeled SM and Cer analogues - which they add to the lipid extraction. However, this is absolutely common practice in mass spectrometric lipidomics studies. While the HPLC settings are described quite extensively, important information on the identification and quantification of SM and Cer are missing. What mass transitions were analyzed for the sphingomyelin and ceramide subspecies? What reference substances were used to confirm the identity of the subspecies? Was a saponification step included during lipid extraction to avoid interference with phosphatidylcholines (PCs)? This is quite crucial since SMs and PCs generate analogous fragment ions by CID, e.g. m/z 184 and 86. Can the authors safely rule out the possibility that overlap of SM and PC isotope signals occurred? For example, precursor ions of C20:0 SM (m/z 759.6) and PC 34:1 (m/z 760.6) differ by only 1 Da. In case of resubmission, authors should answer these open questions in sphingolipid quantification to make the compiled data appear more credible.

  1. No internal standard was used for the quantification.
    2. Identification of 24 SM species was done in MRM on the basis of theoretical parent ion masses. Quantification was done on basis of the calibration curves obtained from three different chain length  standards , and extrapolating to all the other species
    3. This information is reported in Ref. 26-27
    4. No reference substance was used to identify the subspecies
    5. No, a saponification step was not included
    6.  We can only speculate that the chemistry of phenilhexyl column is able to separate SM from PC reducing the possibility of interference (lines 166-169).

Round 2

Reviewer 2 Report

While the authors have satisfactorily addressed the formal issues (readability of the graphs, clarity of the introduction), the methodological inadequacies remain. The manuscript has not improved at all in this respect. Rather, my doubts about the validity of the lipidomics data collected have become more pronounced.

Lipid extraction-based sphingolipidomics analysis by LC-MS/MS without the use of an internal standard of any kind capable of correcting for different extraction efficiencies as well as fluctuations in the analysis is beyond an accurate analytical approach. It is also not comprehensible why this was omitted. There are numerous e.g. deuterated SM standards available for purchase.

The description of the LC-MS/MS methodology also remains inadequate, as no MRMs used are presented despite request.  

Furthermore, no attempt was made to convince the reviewer (e.g. by means of exemplary chromatograms, additional experiments e.g. with vs. w/o saponification) that the method used is actually capable of separating SMs and PCs and that there is therefore no risk of misinterpretation of the data due to overlapping signals. Though, this is an important point, since not more than 4 reference compounds were used for identification of 24 SMs in total.

Instead, the authors speculate that their methodology is probably capable of separating SMs and PCs and that interference is therefore unlikely. This leads me to speculate also whether the data presented is correct or not. Readers of IJMS with expertise in mass spectrometric lipid analysis will feel the same way, which is why I continue to consider this study not suitable for publication.